# Regularizing Trajectory Optimization
# with Denoising Autoencoders

**Rinu Boney**[*]
Aalto University & Curious AI
rinu.boney@aalto.fi

**Norman Di Palo**[*]
Sapienza University of Rome
normandipalo@gmail.com

**Mathias Berglund**
Curious AI

**Alexander Ilin**
Aalto University & Curious AI

**Juho Kannala**
Aalto University

**Antti Rasmus**
Curious AI

**Harri Valpola**
Curious AI

## Abstract

Trajectory optimization using a learned model of the environment is one of the core elements of model-based reinforcement learning. This procedure often suffers from exploiting inaccuracies of the learned model. We propose to regularize trajectory optimization by means of a denoising autoencoder that is trained on the same trajectories as the model of the environment. We show that the proposed regularization leads to improved planning with both gradient-based and gradient-free optimizers. We also demonstrate that using regularized trajectory optimization leads to rapid initial learning in a set of popular motor control tasks, which suggests that the proposed approach can be a useful tool for improving sample efficiency.

## 1 Introduction

State-of-the-art reinforcement learning (RL) often requires a large number of interactions with the environment to learn even relatively simple tasks [11]. It is generally believed that model-based RL can provide better sample-efficiency [9, 2, 5] but showing this in practice has been challenging. In this paper, we propose a way to improve planning in model-based RL and show that it can lead to improved performance and better sample efficiency.

In model-based RL, planning is done by computing the expected result of a sequence of future actions using an explicit model of the environment. Model-based planning has been demonstrated to be efficient in many applications where the model (a simulator) can be built using first principles. For example, model-based control is widely used in robotics and has been used to solve challenging tasks such as human locomotion [34, 35] and dexterous in-hand manipulation [21].

In many applications, however, we often do not have the luxury of an accurate simulator of the environment. Firstly, building even an approximate simulator can be very costly even for processes whose dynamics is well understood. Secondly, it can be challenging to align the state of an existing simulator with the state of the observed process in order to plan. Thirdly, the environment is often non-stationary due to, for example, hardware failures in robotics, change of the input feed and deactivation of materials in industrial process control. Thus, learning the model of the environment is the only viable option in many applications and learning needs to be done for a live system. And since many real-world systems are very complex, we are likely to need powerful function approximators, such as deep neural networks, to learn the dynamics of the environment.

However, planning using a learned (and therefore inaccurate) model of the environment is very difficult in practice. The process of optimizing the sequence of future actions to maximize the

---

[*]Equal contribution, rest in alphabetical order

expected return (which we call trajectory optimization) can easily exploit the inaccuracies of the model and suggest a very unreasonable plan which produces highly over-optimistic predicted rewards. This optimization process works similarly to adversarial attacks [1, 13, 33, 7] where the input of a trained model is modified to achieve the desired output. In fact, a more efficient trajectory optimizer is more likely to fall into this trap. This can arguably be the reason why gradient-based optimization (which is very efficient at for example learning the models) has not been widely used for trajectory optimization.

In this paper, we study this adversarial effect of model-based planning in several environments and show that it poses a problem particularly in high-dimensional control spaces. We also propose to remedy this problem by regularizing trajectory optimization using a denoising autoencoder (DAE) [37]. The DAE is trained to denoise trajectories that appeared in the past experience and in this way the DAE learns the distribution of the collected trajectories. During trajectory optimization, we use the denoising error of the DAE as a regularization term that is subtracted from the maximized objective function. The intuition is that the denoising error will be large for trajectories that are far from the training distribution, signaling that the dynamics model predictions will be less reliable as it has not been trained on such data. Thus, a good trajectory has to give a high predicted return and it can be only moderately novel in the light of past experience.

In the experiments, we demonstrate that the proposed regularization significantly diminishes the adversarial effect of trajectory optimization with learned models. We show that the proposed regularization works well with both gradient-free and gradient-based optimizers (experiments are done with cross-entropy method [3] and Adam [14]) in both open-loop and closed-loop control. We demonstrate that improved trajectory optimization translates to excellent results in early parts of training in standard motor-control tasks and achieve competitive performance after a handful of interactions with the environment.

## 2 Model-Based Reinforcement Learning

In this section, we explain the basic setup of model-based RL and present the notation used. At every time step $t$, the environment is in state $s_t$, the agent performs action $a_t$, receives reward $r_t = r(s_t, a_t)$ and the environment transitions to new state $s_{t+1} = f(s_t, a_t)$. The agent acts based on the observations $o_t = o(s_t)$ which is a function of the environment state. In a fully observable Markov decision process (MDP), the agent observes full state $o_t = s_t$. In a partially observable Markov decision process (POMDP), the observation $o_t$ does not completely reveal $s_t$. The goal of the agent is select actions $\{a_0, a_1, \dots\}$ so as to maximize the return, which is the expected cumulative reward $\mathbb{E}\left[\sum_{t=0}^{\infty} r(s_t, a_t)\right]$.

In the model-based approach, the agent builds the dynamics model of the environment (forward model). For a fully observable environment, the forward model can be a fully-connected neural network trained to predict the state transition from time $t$ to $t + 1$:

$$s_{t+1} = f_\theta(s_t, a_t). \tag{1}$$

In partially observable environments, the forward model can be a recurrent neural network trained to directly predict the future observations based on past observations and actions:

$$o_{t+1} = f_\theta(o_0, a_0, \dots, o_t, a_t). \tag{2}$$

In this paper, we assume access to the reward function and that it can be computed from the agent observations, that is $r_t = r(o_t, a_t)$.

At each time step $t$, the agent uses the learned forward model to plan the sequence of future actions $\{a_t, \dots, a_{t+H}\}$ so as to maximize the expected cumulative future reward.

$$G(a_t, \dots, a_{t+H}) = \mathbb{E}\left[\sum_{\tau=t}^{t+H} r(o_\tau, a_\tau)\right]$$

$$a_t, \dots, a_{t+H} = \arg\max G(a_t, \dots, a_{t+H}).$$

This process is called trajectory optimization. The agent uses the learned model of the environment to compute the objective function $G(a_t, \dots, a_{t+H})$. The model (1) or (2) is unrolled $H$ steps into the future using the current plan $\{a_t, \dots, a_{t+H}\}$.

---

**Algorithm 1** End-to-end model-based reinforcement learning

---

    Collect data $\mathbb{D}$ by random policy.
    **for** each episode **do**
        Train dynamics model $f_\theta$ using $\mathbb{D}$.
        **for** time $t$ until the episode is over **do**
            Optimize trajectory $\{a_t, o_{t+1}, \ldots, a_{t+H}, o_{t+H+1}\}$.
            Implement the first action $a_t$ and get new observation $o_t$.
        **end for**
        Add data $\{(s_1, a_1, \ldots, a_T, o_T)\}$ from the last episode to $\mathbb{D}$.
    **end for**

---

The optimized sequence of actions from trajectory optimization can be directly applied to the environment (open-loop control). It can also be provided as suggestions to a human operator with the possibility for the human to change the plan (human-in-the-loop). Open-loop control is challenging because the dynamics model has to be able to make accurate long-range predictions. An approach which works better in practice is to take only the first action of the optimized trajectory and then re-plan at each step (closed-loop control). Thus, in closed-loop control, we account for possible modeling errors and feedback from the environment. In the control literature, this flavor of model-based RL is called model-predictive control (MPC) [22, 30, 16, 24].

The typical sequence of steps performed in model-based RL are: 1) collect data, 2) train the forward model $f_\theta$, 3) interact with the environment using MPC (this involves trajectory optimization in every time step), 4) store the data collected during the last interaction and continue to step 2. The algorithm is outlined in Algorithm 1.

## 3 Regularized Trajectory Optimization

### 3.1 Problem with using learned models for planning

In this paper, we focus on the inner loop of model-based RL which is trajectory optimization using a *learned* forward model $f_\theta$. Potential inaccuracies of the trained model cause substantial difficulties for the planning process. Rather than optimizing what really happens, planning can easily end up exploiting the weaknesses of the predictive model. Planning is effectively an adversarial attack against the agent's own forward model. This results in a wide gap between expectations based on the model and what actually happens.

We demonstrate this problem using a simple industrial process control benchmark from [28]. The problem is to control a continuous nonlinear reactor by manipulating three valves which control flows in two feeds and one output stream. Further details of the process and the control problem are given in Appendix A. The task considered in [28] is to change the product rate of the process from 100 to 130 kmol/h. Fig. 1a shows how this task can be performed using a set of PI controllers proposed in [28]. We trained a forward model of the process using a recurrent neural network (2) and the data collected by implementing the PI control strategy for a set of randomly generated targets. Then we optimized the trajectory for the considered task using gradient-based optimization, which produced results in Fig. 1b. One can see that the proposed control signals are changed abruptly and the trajectory imagined by the model significantly deviates from reality. For example, the pressure constraint (of max 300 kPa) is violated. This example demonstrates how planning can easily exploit the weaknesses of the predictive model.

### 3.2 Regularizing Trajectory Optimization with Denoising Autoencoders

We propose to regularize the trajectory optimization with denoising autoencoders (DAE). The idea is that we want to reward familiar trajectories and penalize unfamiliar ones because the model is likely to make larger errors for the unfamiliar ones.

This can be achieved by adding a regularization term to the objective function:

$$G_{\text{reg}} = G + \alpha \log p(o_t, a_t \ldots, o_{t+H}, a_{t+H}), \tag{3}$$

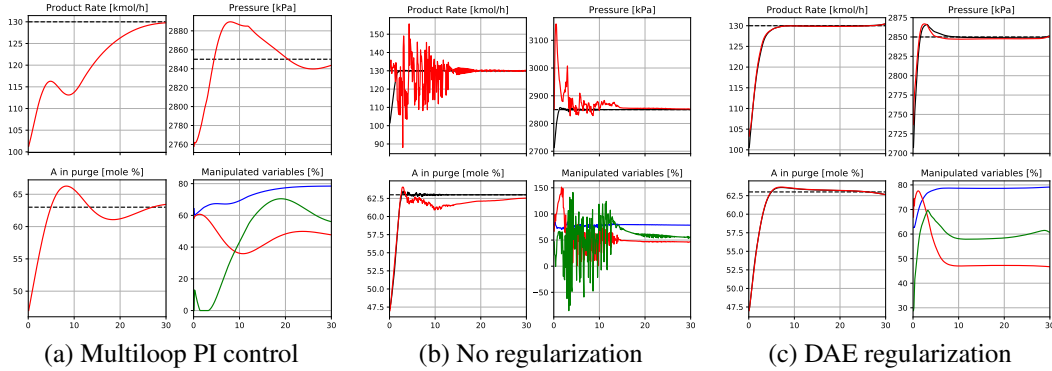

|  (a) Multiloop PI control | (b) No regularization | (c) DAE regularization |

Figure 1: Open-loop planning for a continuous nonlinear two-phase reactor from [28]. Three subplots in every subfigure show three measured variables (solid lines): product rate, pressure and A in the purge. The black curves represent the model's imagination while the red curves represent the reality if those controls are applied in an open-loop mode. The targets for the variables are shown with dashed lines. The fourth (low right) subplots show the three manipulated variables: valve for feed 1 (blue), valve for feed 2 (red) and valve for stream 3 (green).

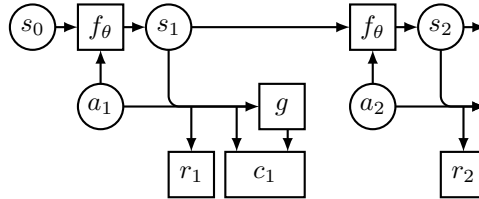

Figure 2: Example: fragment of a computational graph used during trajectory optimization in an MDP. Here, window size $w = 1$, that is the DAE penalty term is $c_1 = \|g([s_1, a_1]) - [s_1, a_1]\|^2$.

where $p(o_t, a_t, \ldots, o_{t+H}, a_{t+H})$ is the probability of observing a given trajectory in the past experience and $\alpha$ is a tuning hyperparameter. In practice, instead of using the joint probability of the whole trajectory, we use marginal probabilities over short windows of size $w$:

$$G_{\text{reg}} = G + \alpha \sum_{\tau=t}^{t+H-w} \log p(x_\tau) \tag{4}$$

where $x_\tau = \{o_\tau, a_\tau, \ldots o_{\tau+w}, a_{\tau+w}\}$ is a short window of the optimized trajectory.

Suppose we want to find the optimal sequence of actions by maximizing (4) with a gradient-based optimization procedure. We can compute gradients $\frac{\partial G_{\text{reg}}}{\partial a_i}$ by backpropagation in a computational graph where the trained forward model is unrolled into the future (see Fig. 2). In such backpropagation-through-time procedure, one needs to compute the gradient with respect to actions $a_i$.

$$\frac{\partial G_{\text{reg}}}{\partial a_i} = \frac{\partial G}{\partial a_i} + \alpha \sum_{\tau=i}^{i+w} \frac{\partial x_\tau}{\partial a_i} \frac{\partial}{\partial x_\tau} \log p(x_\tau), \tag{5}$$

where we denote by $x_\tau$ a concatenated vector of observations $o_\tau, \ldots o_{\tau+w}$ and actions $a_\tau, \ldots a_{\tau+w}$, over a window of size $w$. Thus to enable a regularized gradient-based optimization procedure, we need means to compute $\frac{\partial}{\partial x_\tau} \log p(x_\tau)$.

In order to evaluate $\log p(x_\tau)$ (or its derivative), one needs to train a separate model $p(x_\tau)$ of the past experience, which is the task of unsupervised learning. In principle, any probabilistic model can be used for that. In this paper, we propose to regularize trajectory optimization with a denoising autoencoder (DAE) which does not build an explicit probabilistic model $p(x_\tau)$ but rather learns to approximate the derivative of the log probability density. The theory of denoising [23, 27] states that

the optimal denoising function $g(\tilde{x})$ (for zero-mean Gaussian corruption) is given by:

$$g(\tilde{x}) = \tilde{x} + \sigma_n^2 \frac{\partial}{\partial \tilde{x}} \log p(\tilde{x}),$$

where $p(\tilde{x})$ is the probability density function for data $\tilde{x}$ corrupted with noise and $\sigma_n$ is the standard deviation of the Gaussian corruption. Thus, the DAE-denoised signal minus the original gives the gradient of the log-probability of the data distribution convolved with a Gaussian distribution: $\frac{\partial}{\partial \tilde{x}} \log p(\tilde{x}) \propto g(x) - x$. Assuming $\frac{\partial}{\partial \tilde{x}} \log p(\tilde{x}) \approx \frac{\partial}{\partial x} \log p(x)$ yields

$$\frac{\partial G_{\text{reg}}}{\partial a_i} = \frac{\partial G}{\partial a_i} + \alpha \sum_{\tau=i}^{i+w} \frac{\partial x_\tau}{\partial a_i}(g(x_\tau) - x_\tau). \qquad (6)$$

Using $\frac{\partial}{\partial \tilde{x}} \log p(\tilde{x})$ instead of $\frac{\partial}{\partial x} \log p(x)$ can behave better in practice because it is similar to replacing $p(x)$ with its Parzen window estimate [36]. In automatic differentiation software, this gradient can be computed by adding the penalty term $\|g(x_\tau) - x_\tau\|^2$ to $G$ and stopping the gradient propagation through $g$. In practice, stopping the gradient through $g$ did not yield any benefits in our experiments compared to simply adding the penalty term $\|g(x_\tau) - x_\tau\|^2$ to the cumulative reward, so we used the simple penalty term in our experiments. Also, this kind of regularization can easily be used with gradient-free optimization methods such as cross-entropy method (CEM) [3].

Our goal is to tackle high-dimensional problems and expressive models of dynamics. Neural networks tend to fare better than many other techniques in modeling high-dimensional distributions. However, using a neural network or any other flexible parameterized model to estimate the input distribution poses a dilemma: the regularizing network which is supposed to keep planning from exploiting the inaccuracies of the dynamics model will itself have weaknesses which planning will then exploit. Clearly, DAE will also have inaccuracies but planning will not exploit them because unlike most other density models, DAE develops an explicit model of the gradient of logarithmic probability density.

The effect of adding DAE regularization in the industrial process control benchmark discussed in the previous section is shown in Fig. 1c.

### 3.3 Related work

Several methods have been proposed for planning with learned dynamics models. Locally linear time-varying models [17, 19] and Gaussian processes [8, 15] or mixture of Gaussians [29] are data-efficient but have problems scaling to high-dimensional environments. Recently, deep neural networks have been successfully applied to model-based RL. Nagabandi et al. [24] use deep neural networks as dynamics models in model-predictive control to achieve good performance, and then shows how model-based RL can be fine-tuned with a model-free approach to achieve even better performance. Chua et al. [5] introduce PETS, a method to improve model-based performance by estimating and propagating uncertainty with an ensemble of networks and sampling techniques. They demonstrate how their approach can beat several recent model-based and model-free techniques. Clavera et al. [6] combines model-based RL and meta-learning with MB-MPO, training a policy to quickly adapt to slightly different learned dynamics models, thus enabling faster learning.

Levine and Koltun [20] and Kumar et al. [17] use a KL divergence penalty between action distributions to stay close to the training distribution. Similar bounds are also used to stabilize training of policy gradient methods [31, 32]. While such a KL penalty bounds the evolution of action distributions, the proposed method also bounds the familiarity of states, which could be important in high-dimensional state spaces. While penalizing unfamiliar states also penalize exploration, it allows for more controlled and efficient exploration. Exploration is out of the scope of the paper but was studied in [10], where a non-zero optimum of the proposed DAE penalty was used as an intrinsic reward to alternate between familiarity and exploration.

## 4 Experiments on Motor Control

We show the effect of the proposed regularization for control in standard Mujoco environments: Cartpole, Reacher, Pusher, Half-cheetah and Ant available in [4]. See the description of the environments in Appendix B. We use the Probabilistic Ensembles with Trajectory Sampling (PETS) model from [5] as the baseline, which achieves the best reported results on all the considered tasks except for Ant.

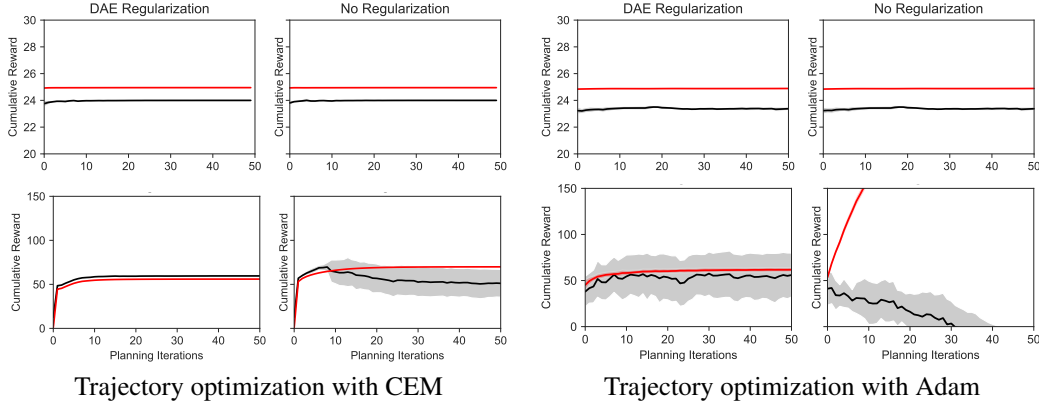

Trajectory optimization with CEM          Trajectory optimization with Adam

Figure 3: Visualization of trajectory optimization at timestep $t = 50$. Each row has the same model but a different optimization method. The models are obtained by 5 episodes of end-to-end training. Row above: Cartpole environment. Row below: Half-cheetah environment. Here, the red lines denote the rewards predicted by the model (imagination) and the black lines denote the true rewards obtained when applying the sequence of optimized actions (reality). For a low-dimensional action space (Cartpole), trajectory optimizers do not exploit inaccuracies of the dynamics model and hence DAE regularization does not affect the performance noticeably. For a higher-dimensional action space (Half-cheetah), gradient-based optimization without any regularization easily exploits inaccuracies of the dynamics model but DAE regularization is able to prevent this. The effect is less pronounced with gradient-free optimization but still noticeable.

The PETS model consists of an ensemble of probabilistic neural networks and uses particle-based trajectory sampling to regularize trajectory optimization. We re-implemented the PETS model using the code provided by the authors as a reference.

## 4.1 Regularized trajectory optimization with models trained with PETS

In MPC, the innermost loop is open-loop control which is then turned to closed-loop control by taking in new observations and replanning after each action. Fig. 3 illustrates the adversarial effect during open-loop trajectory optimization and how DAE regularization mitigates it. In Cartpole environment, the learned model is very good already after a few episodes of data and trajectory optimization stays within the data distribution. As there is no problem to begin with, regularization does not improve the results. In Half-cheetah environment, trajectory optimization manages to exploit the inaccuracies of the model which is particularly apparent in gradient-based Adam. DAE regularization improves both but the effect is much stronger with Adam.

The problem is exacerbated in closed-loop control since it continues optimization from the solution achieved in the previous time step, effectively iterating more per action. We demonstrate how regularization can improve closed-loop trajectory optimization in the Half-cheetah environment. We first train three PETS models for 300 episodes using the best hyperparameters reported in [5]. We then evaluate the performance of the three models on five episodes using four different trajectory optimizers: 1) Cross-entropy method (CEM) which was used during training of the PETS models, 2) Adam, 3) CEM with the DAE regularization and 4) Adam with the DAE regularization. The results averaged across the three models and the five episodes are presented in Table 1.

Table 1: Comparison of PETS with CEM and Adam optimizers in Half-cheetah

| Optimizer | CEM | CEM + DAE | Adam | Adam + DAE |
|---|---|---|---|---|
| Average Return | $10955 \pm 2865$ | $12967 \pm 3216$ | – | $12796 \pm 2716$ |

We first note that planning with Adam fails completely without regularization: the proposed actions lead to unstable states of the simulator. Using Adam with the DAE regularization fixes this problem and the obtained results are better than the CEM method originally used in PETS. CEM appears

to regularize trajectory optimization but not as efficiently CEM+DAE. These open-loop results are consistent with the closed-loop results in Fig. 3.

## 4.2 End-to-end training with regularized trajectory optimization

In the following experiments, we study the performance of end-to-end training with different trajectory optimizers used during training. Our agent learns according to the algorithm presented in Algorithm 1. Since the environments are fully observable, we use a feedforward neural network as in (1) to model the dynamics of the environment. Unlike PETS, we did not use an ensemble of probabilistic networks as the forward model. We use a single probabilistic network which predicts the mean and variance of the next state (assuming a Gaussian distribution) given the current state and action. Although we only use the mean prediction, we found that also training to predict the variance improves the stability of the training.

For all environments, we use a dynamics model with the same architecture: three hidden layers of size 200 with the Swish non-linearity [26]. Similar to prior works, we train the dynamics model to predict the difference between $s_{t+1}$ and $s_t$ instead of predicting $s_{t+1}$ directly. We train the dynamics model for 100 or more epochs (see Appendix C) after every episode. This is a larger number of updates compared to five epochs used in [5]. We found that an increased number of updates has a large effect on the performance for a single probabilistic model and not so large effect for the ensemble of models used in PETS. This effect is shown in Fig. 6.

For the denoising autoencoder, we use the same architecture as the dynamics model. The state-action pairs in the past episodes were corrupted with zero-mean Gaussian noise and the DAE was trained to denoise it. Important hyperparameters used in our experiments are reported in the Appendix C. For DAE-regularized trajectory optimization we used either CEM or Adam as optimizers.

The learning progress of the compared algorithms is presented in Fig. 4. Note that we report the *average* returns across different seeds, not the maximum return seen so far as was done in [5].[2] In Cartpole, all the methods converge to the maximum cumulative reward but the proposed method converges the fastest. In the Cartpole environment, we also compare to a method which uses Gaussian Processes (GP) as the dynamics model (algorithm denoted GP-E in [5], which considers only the expectation of the next state prediction). The implementation of the GP algorithm was obtained from the code provided by [5]. Interestingly, our algorithm also surpasses the Gaussian Process (GP) baseline, which is known to be a sample-efficient method widely used for control of simple systems. In Reacher, the proposed method converges to the same asymptotic performance as PETS, but faster. In Pusher, all algorithms perform similarly.

In Half-cheetah and Ant, the proposed method shows very good sample efficiency and very rapid initial learning. The agent learns an effective running gait in only a couple of episodes.[3] The results demonstrate that denoising regularization is effective for both gradient-free and gradient-based planning, with gradient-based planning performing the best. The proposed algorithm learns faster than PETS in the initial phase of training. It also achieves performance that is competitive with popular model-free algorithms such as DDPG, as reported in [5].

However, the performance of the proposed method does not improve after initial 10 episodes, so it does not reach the asymptotic performance of PETS (see results for PETS for Half-cheetah after 300 episodes in Table 1). This result is evidence of the importance of exploration: the DAE regularization essentially penalizes exploration, which can harm asymptotic performance in complex environments. In PETS, CEM leaves some noise in the trajectories, which might help to obtain better asymptotic performance. The result presented in Appendix E provides some evidence that at least a part of the problem is lack of exploration.

We also compare the performance of our method with Model-Based Meta Policy Optimization (MB-MPO) [6], an approach that combines the benefits of model-based RL and meta learning: the algorithm trains a policy using simulations generated by an ensemble of models, learned from data. Meta-learning allows this policy to quickly adapt to the various dynamics, hence learning how to quickly adapt in the real environment, using Model-Agnostic Meta Learning (MAML) [12]. In

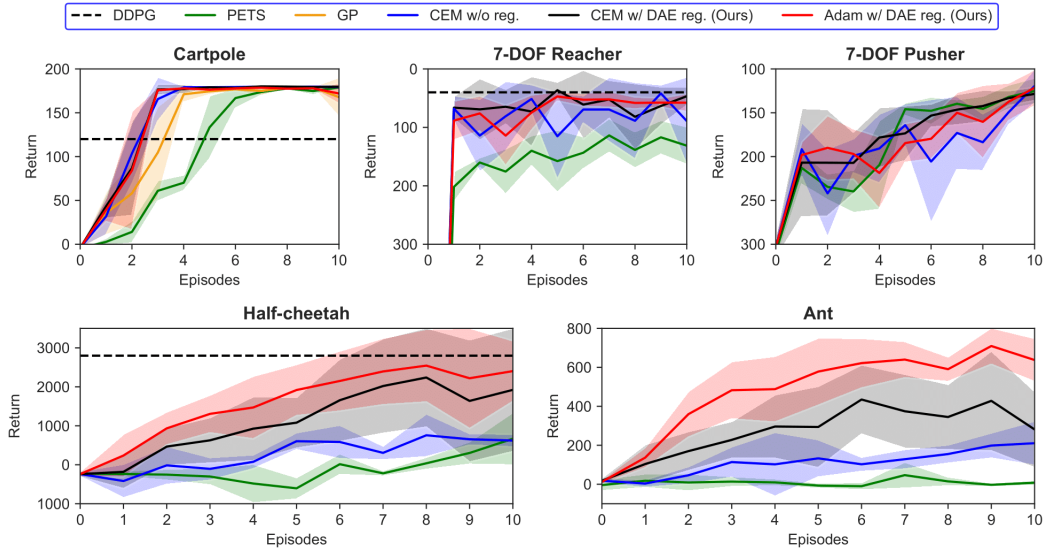

Figure 4: Results of our experiments on the five benchmark environments, in comparison to PETS [5]. We show the return obtained in each episode. All the results are averaged across 5 seeds, with the shaded area representing standard deviation. PETS is a recent state-of-the-art model-based RL algorithm and GP-based (Gaussian Processes) control algorithms are well known to be sample-efficient and are extensively used for the control of simple systems.

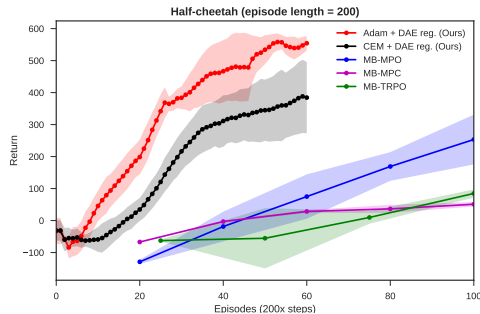

Figure 5: Comparison to MB-MPO [6], MB-TRPO [18] and MB-MPC [24] on Half-cheetah. We plot the average return over the last 20 episodes. Our results are averaged across 3 seeds, with the shaded area representing standard deviation. Note that the comparison numbers are picked from [6] and the results from the first 20 episodes are not reported.

Fig. 5 we compare our method to MB-MPO and other model-based methods included in [6]. This experiment is done in the Half-cheetah environment with shorter episodes (200 timesteps) in order to compare to the results reported in [6]. The results show that our method learns faster than MB-MPO.

## 5 Discussion

In recent years, a lot of effort has been put in making deep reinforcement algorithms more sample-efficient, and thus adaptable to real world scenarios. Model-based reinforcement learning has shown promising results, obtaining sample-efficiency even orders of magnitude better than model-free counterparts, but these methods have often suffered from sub-optimal performance due to many reasons. As already noted in the recent literature [24, 5], out-of-distribution errors and model overfitting are often sources of performance degradation when using complex function approximators. In this work we demonstrated how to tackle this problem using regularized trajectory optimization. Our experiments demonstrate that the proposed solution can improve the performance of model-based reinforcement learning.

While trajectory optimization is a key component in model-based RL, there are clearly several other issues which need to be tackled in complex environments:

- Local minima for trajectory optimization. There can be multiple trajectories that are reasonable solutions but in-between trajectories can be very bad. For example, we can take a step with a right or left foot but both will not work. We tackled this issue by trying multiple initializations, which worked for the considered environments, but better techniques will be needed for more complex environments.

- The planning horizon problem. In the presented experiments, the planning procedure did not care about what happens after the planning horizon. This was not a problem for the considered environments due to nicely formatted reward. Other solutions like value functions, multiple time scales or hierarchy for planning are required with sparser reward problems. All of these are compatible with model-based RL.

- Open-loop vs. closed-loop (compounding errors). The implicit planning assumption of trajectory optimization is open-loop control. However, MPC only takes the first action and then replans (closed-loop control). If the outcome is uncertain (e.g., due to stochastic environments or imperfect forward model), this can lead to overly pessimistic controls.

- Local optima of the policy. This is the well-known exploration-exploitation dilemma. If the model has never seen data of alternative trajectories, it may predict their consequences incorrectly and never try them (because in-between trajectories can be genuinely worse). Good trajectory optimization (exploitation) can harm long-term performance because it reduces exploration, but we believe that it is better to add explicit exploration. With model-based RL, intrinsically motivated exploration is a particularly interesting option because it is possible to balance exploration and the expected cost. This is particularly important in hazardous environments where safe exploration is needed.

- High-dimensional input space. Sensory systems like cameras, lidars and microphones can produce vast amounts of data and it is infeasible to plan based on detailed prediction on low level such as pixels. Also, predictive models of pixels may miss the relevant state.

- Changing environments. All the considered environments were static but real-world systems keep changing. Online learning and similar techniques are needed to keep track of the changing environment.

Still, model-based RL is an attractive approach and not only due to its sample-efficiency. Compared to model-free approaches, model-based learning makes safe exploration and adding known constraints or first-principles models much easier. We believe that the proposed method can be a viable solution for real-world control tasks especially where safe exploration is of high importance.

We are currently working on applying the proposed methods for real-world problems such as assisting operators of complex industrial processes and for control of autonomous mobile machines.

### Acknowledgments

We would like to thank Jussi Sainio, Jari Rosti and Isabeau Prémont-Schwarz for their valuable contributions in the experiments on industrial process control.

## Footnotes

[2] Because of the different metric used, the PETS results presented in this paper may appear worse than in [5]. However, we verified that our implementation of PETS obtains similar results to [5] for the metric used in [5].

[3] Videos of our agents during training can be found at `https://sites.google.com/view/regularizing-mbrl-with-dae/home`.

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
