[Supplementary Material]

# A Industrial Process Control Benchmark

To study trajectory optimization, we first consider the problem of control of a simple industrial process. An effective industrial control system could achieve better production and economic efficiency than manually operated controls. In this paper, we learn the dynamics of an industrial process and use it to optimize the controls, by minimizing a cost function. In some critical processes, safety is of utmost importance and regularization methods could prevent adaptive control methods from exploring unsafe trajectories.

We consider the problem of control of a continuous nonlinear two-phase reactor from [28]. The simulated industrial process consists of a single vessel that represents a combination of the reactor and separation system. The process has two feeds: one contains substances A, B and C and the other one is pure A. Reaction $A + C \rightarrow D$ occurs in the vapour phase. The liquid is pure D which is the product. The process is manipulated by three valves which regulate the flows in the two feeds and an output stream which contains A, B and C. The plant has ten measured variables including the flow rates of the four streams ($F_1, \ldots, F_4$), pressure, liquid holdup volume and mole % of A, B and C in the purge. The control problem is to transition to a specified product rate and maintain it by manipulating the three valves. The pressure must be kept below the shutdown limit of 3000 kPa. The original paper suggests a multiloop control strategy with several PI controllers [28].

We collected simulated data corresponding to about 0.5M steps of operation by randomly generating control setpoints and using the original multiloop control strategy. The collected data were used to train a neural network model with one layer of 80 LSTM units and a linear readout layer to predict the next-step measurements. The inputs were the three controls and the ten process measurements. The data were pre-processed by scaling such that the standard deviation of the derivatives of each measured variable was of the same scale. This way, the model learned better the dynamics of slow changing variables. We used a fully-connected network architecture with 8 hidden layers (100-200-100-20-100-200-100) to train a DAE on windows of five successive measurement-control pairs. The scaled measurement-control pairs in a window were concatenated to a single vector and corrupted with zero-mean Gaussian noise ($\sigma = 0.03$) and the DAE was trained to denoise it.

The trained model was then used for optimizing a sequence of actions to ramp production as rapidly as possible from $F_4 = 100$ to $F_4 = 130$ kmol h$^{-1}$, while satisfying all other constraints [Scenario II from 28]. We formulated the objective function as the Euclidean distance to the desired targets (after pre-processing). The targets corresponded to the following targets for three measurements: $F_4 = 130$ kmol h$^{-1}$ for product rate, 2850 kPa for pressure and 63 mole % for A in the purge.

We optimized a plan of actions 30 hours ahead (or 300 discretized time steps). The optimized sequence of controls were initialized with the original multiloop policy applied to the trained dynamics model. That control sequence together with the predicted and the real outcomes (black and red curves respectively) are shown in Fig. 1a. We then optimized the control sequence using 10000 iterations of Adam with learning rate 0.01 without and with DAE regularization (with penalty $\alpha \| g(x_t) - x_t \|^2$).

The results are shown in Fig. 1. One can see that without regularization the control signals are changed abruptly and the trajectory imagined by the model deviates from reality (Fig. 1b). In contrast, the open-loop plan found with the DAE regularization is noticeably the best solution (Fig. 1c), leading the plant to the specified product rate much faster than the human-engineered multiloop PI control from [28]. The imagined trajectory (black) stays close to predictions and the targets are reached in about ten hours. This shows that even in a low-dimensional environment with a large amount of training data, regularization is necessary for planning using a learned model.

# B Description of Environments

**Cartpole**. This task involves a pole attached to a moving cart in a frictionless track, with the goal of swinging up the pole and balancing it in an upright position in the center of the screen. The cost at every time step is measured as the angular distance between the tip of the pole and the target position. Each episode is 200 steps long.

**Reacher**. This environment consists of a simulated PR2 robot arm with seven degrees of freedom, with the goal of reaching a particular position in space. The cost at every time step is measured as the

distance between the arm and the target position. The target position changes every episode. Each episode is 150 steps long.

**Pusher**. This environment also consists of a simulated PR2 robot arm, with a goal of pushing an object to a target position that changes every episode. The cost at every time step is measured as the distance between the object and the target position. Each episode is 150 steps long.

**Half-cheetah**. This environment involves training a two-legged "half-cheetah" to run forward as fast as possible by applying torques to 6 different joints. The cost at every time step is measured as the negative forward velocity. Each episode is 1000 steps long, but the length is reduced to 200 for the benchmark with [6].

**Ant**. This is the most challenging environment we consider. It consists of a four-legged "ant" controlled by applying torques to its 8 joints. Similar to [25], we use a gear ratio to 30 for all joints (this prevents the ant from flipping over frequently during the initially phase of training). The cost, similar to Half-cheetah, is the negative forward velocity. Each episode is 1000 steps long.

Table 2: Dimensionalities of observation and action spaces of the environments used in this paper

| Environment | Observation space | Action space |
|---|---|---|
| Cartpole | 5 | 1 |
| Reacher | 17 | 7 |
| Pusher | 20 | 7 |
| Half-cheetah | 19 | 6 |
| Ant | 111 | 8 |

## C  Additional Experimental Details

For MPC, we use the same planning horizon as PETS (Table 5). The important hyperparameters for all our experiments are shown in Tables 3 and 4. We found the DAE noise level, regularization penalty weight $\alpha$ and Adam learning rate to be the most important hyperparameters.

Figure 6: Effect of increased number of training epochs after every episode: we can see that training the dynamics model for more epochs after each episode leads to a much better performance in the initial episodes. With this modification, a single dynamics model with no regularization seems to work almost as well as PETS. It can also be clearly seen that the use of denoising regularization enables an improvement in the learning progress. To compare with PETS, we used the CEM optimizer in this ablation study.

Table 3: Important hyperparameters used in our experiments for comparison with PETS. Additionally, for the experiments with gradient-based trajectory optimization on Reacher and Pusher, we initialize the trajectory with a few iterations (2 iterations for Reacher and 5 iterations for Pusher) of CEM.

| Environment | Optimizer | Optim Iters | Epochs | Adam LR | $\alpha$ | DAE noise $\sigma$ |
|---|---|---|---|---|---|---|
| Cartpole | CEM | 5 | 500 | - | 0.001 | 0.1 |
| | Adam | 10 | 500 | 0.001 | 0.001 | 0.2 |
| Reacher | CEM | 5 | 500 | - | 0.01 | 0.1 |
| | Adam | 5 | 300 | 1 | 0.01 | 0.1 |
| Pusher | CEM | 5 | 500 | - | 0.01 | 0.1 |
| | Adam | 5 | 300 | 1 | 0.01 | 0.1 |
| Half-cheetah | CEM | 5 | 100 | - | 2 | 0.1 |
| | Adam | 10 | 200 | 0.1 | 1 | 0.2 |
| Ant | CEM | 5 | 400 | - | 0.045 | 0.3 |
| | Adam | 10 | 1000 | 0.075 | 0.03 | 0.4 |

Table 4: Important hyperparameters used in our experiments for comparison with MB-MPO

| Environment | Optimizer | Optim Iters | Epochs | Adam LR | $\alpha$ | DAE noise $\sigma$ |
|---|---|---|---|---|---|---|
| Half-cheetah | CEM | 5 | 20 | - | 2 | 0.2 |
| | Adam | 10 | 40 | 0.1 | 1 | 0.1 |

Table 5: MPC planning horizons used in our experiments

| Environment | Cartpole | Reacher | Pusher | Half-cheetah | Ant |
|---|---|---|---|---|---|
| Planning Horizon | 25 | 25 | 25 | 30 | 35 |

## D  Comparison to Gaussian regularization

To emphasize the importance of denoising regularization, we also compare against a simple Gaussian regularization baseline: we fit a Gaussian distribution (with diagonal covariance matrix) to the states and actions in the replay buffer and regularize the trajectory optimization by adding a penalty term to the cost, proportional to the negative log probability of the states and actions in the trajectory (Equation 4). The performance of this baseline in the Half-cheetah task (with an episode length of 200) is shown in Fig. 7. We observe that the Gaussian distribution poorly fits the trajectories and consistently leads the optimization to a bad local minimum.

## E  Preliminary Experiments on Exploration

To improve the asymptotic performance of our agent, we perform some preliminary experiments on exploration by injecting random noise into the optimized actions. In Figure 8, we show that asymptotic performance can greatly benefit from random exploration, suggesting a line of future work.

## F  Visualization of Trajectory Optimization in End-to-End Experiments

In Figures 9 and 10, we visualize trajectory optimization at different timesteps $t$ during Episode 5 of end-to-end experiments in Cartpole and Half-cheetah. It can be observed that the DAE penalty correlates with the inaccuracies of the model and that the DAE regularization is effective in guiding the optimization procedure to remain within the data distribution.

**Half-cheetah (episode length = 200)**

Figure 7: Comparison to Gaussian regularization: we can see that trajectory optimization with Adam without any regularization is very unstable and completely fails in the initial episodes. While Gaussian regularization helps in the first few episodes, it is not able to fit the data properly and seems to consistently lead the optimization to a local minimum. As shown earlier in Fig. 5, denoising regularization is able to successfully regularize the optimization, enabling good asymptotic performance from very few episodes of interaction.

**Half-cheetah**

Figure 8: In this plot we show the cumulative reward obtained during training by our method when we inject noise to actions in order to improve exploration of the state-action space. Plots are averaged over 5 seeds, and show mean and standard deviation.

Figure 9: Visualization of trajectory optimization at different timesteps $t$ during Episode 5 of end-to-end training in the Cartpole environment. Here, the red line denotes the rewards predicted by the model (imagination) and the black line denotes the true rewards obtained when applying the sequence of optimized actions (reality).

Figure 10: Visualization of trajectory optimization at different timesteps $t$ during Episode 5 of end-to-end training in the Half-cheetah environment. Here, the red line denotes the rewards predicted by the model (imagination) and the black line denotes the true rewards obtained when applying the sequence of optimized actions (reality).