[Reviews · NeurIPS 2019]

Reviewer 1



The paper addresses the problem of reducing the exploitation of inaccuracies of learned dynamics models by trajectory optimization algorithms in model-based Reinforcement Learning. For this, it proposes to add a regularizer to the optimization cost which writes as an estimation of the log probability (in a local window) of sampling the optimized trajectory from the distribution of known trajectories. The idea is to avoid trajectories deviating too much from the data used to learn the dynamics model, and hence avoid unreliable solutions. The authors propose to estimate the log probability term with a denoising autoencoder network. They provide multiple experiments comparing their method to other state-of-the-art approaches on known environments/datasets. The paper is really well organized and well written, which makes it easy to read and understand.Nonetheless, I found confusing/misleading to state in line 161 that exploration is out of the scope of the paper. Indeed, in my understanding, the motivation for the regularization proposed is to penalize exploration of unfamiliar parts of the state-control domain and in that sense it has everything to do with exploration. Similarly, you say in line 260 that your method is a good solution for safe exploration in real-world problems while system safety (i.e. preventing the system from entering a state in which it will break) is never really considered in the article. What is really considered is the reliability of the learned dynamics model along the proposed trajectory. Indeed, the method proposed penalizes exploration so that new trajectories don’t go too far from old known ones, and by iterating on this process of regularized optimization —> gathering data along optimized trajectories —> updating estimation of trajectories distributions —> re-optimizing new regularized trajectories, it seems that it is possible to obtain unsafe trajectories at some point if no other assumptions are made. Other clarity related minor remarks are listed below: - What are the gray shades in figure 3 ? Not in the caption nor text. - Line 192: as efficiently as - Line 192-193: it seems that you mis-switched open-loop and closed-loop Concerning the methodological part, the idea of regularizing the optimization using the density of real trajectories used to train the dynamics model is not new. As pointed in the paper itself (references [17, 20, 30, 31]), KL-divergence based regularizers have already been proposed in the model-based RL literature. Also, the use of local density estimators trained on past experience to penalize unreliable trajectories is a known technique in applied optimal control (see for example Gaussian Mixture Penalty for Trajectory Optimization Problems, 2018). I believe that further comments motivating the choice of a DAE could be useful for the reader considering that DAE only estimates the derivative of the log likelihood. Indeed, variational autoencoders or other more standards density estimation models like Gaussian Mixture models or Kernel density estimators seem like a more natural choice. One potential advantage of deep network models such as DAE or VAE could be that they are less sensitive to dimensionality curse than standard methods. Indeed, it is said in the article that high-dimensional control spaces are the focus of the study and that adversarial effects are more frequent in such context. However, the dimensions of the problems considered in the experimental part are not mentioned. I believe that this information is crucial to better understand the strengths of the proposed approach. Besides this, I must say that the experimental part of the paper is of great quality given the large number of experiments considered, the fairly complex control tasks tackled and a comparison which seems fair to some state-of-the-art techniques. I wonder however why the authors chose to compare the proposed method to CEM instead of a more standard trajectory optimization technique, such as dynamic programming. Also, concerning figure 4, we can see on the results of the Pusher, the Half-cheetah and the Ant that none of the returns have stabilized after 10 episodes and it seems natural to wonder what happens next. There is indeed a comment on that in line 230, saying that the proposed method does not reach the asymptotic performance of PETS, with more interesting results in figure 8 of the supplementary material showing that the proposed method regularizes exploration too much and does not beat PETS with noise injection after 50 episodes. Despite the honest comment on this limitation in the main document, I believe that the results from figure 8 are too interesting to be left to supplementary material. On the contrary, concerning footnote 1 in page 7, I think that the results of the verifications carried on the re-implementation of PETS should be added to the supplemental material, as they seem important for reproducibility. Despite the clarity of the paper and the great quality of the experiments presented therein, it appears to be a fairly straightforward application of existing techniques with new models (RNN, DAE). As such, I think it would be well-suited for publication in a more applied journal/conference or maybe in a NIPS workshop. -- I have read the authors response. Although they have addressed most of my concerns, I still think that experimental results are needed to justify their claims.

Reviewer 2



The authors begin with a survey of the related work which is a good representative of the current work in this area. I like their first process control experiment as it clearly lays out what is being achieved, with respect to conventional state of the art (i.e., not just the last few years' DRL papers). I would like to have better understood how well the proposed algorithms would fare by also looking at a baseline with the true models. This would tell us not just the improvements from below, but the upper envelopes in this setup. I find that the experiments have explored the factors of variations suitably, within the confines of the environment within which they are setup. One major source of model uncertainty, in the real world, is uncertainty in the environment (both epistemic and aleatory). It is not yet clear to me how this method could cope with that, and that would be a helpful experiment.

Reviewer 3



Originality: The denoising autoencoder regularized trajectory optimization is a novel method to deal with the adversarial effect with the learned dynamic model in the model-based RL framework. Instead of investigating new ways to learn the dynamic model, this paper tries to improve the performance and sample efficiency from the trajectory optimization regularization point of view, which is interesting and nice try. Quality: The quality of the paper is good. The only concern I have is about the experimental evaluations. The chosen tasks are relatively simple, the implementation of the more complex experiments are expected. Also, a comparison to more algorithms would be useful. The method of [Ha & Schmidhuber] is an important work in the field of model-based RL, the comparison to world model is encouraged to include. I understand that the problem set up in the current submission is different to world model. However, the dynamic model learning method should also affect the performance, which is not discussed in the paper. Thus, the model learning method in [Ha & Schmidhuber] (VAE-based model) or some other methods, is encouraged to try in the implementation of your model learning part. [Ha & Schmidhuber] David Ha and Jurgen Schmidhuber. World models. arXiv: 1803.10122, 2018. Clarity: The paper is well written and organized, and it is easy to understand. Significance: The applicability of model-based RL is on the premise that the dynamic of environment is learned accurately. However, we all know that the approximation of the environment is challenging especially in the complex real-world problem. This paper tries to mitigate this bottleneck problem of model-based RL methods from a different view, which is with great significance. It is nice to see more real-world applications of the proposed method. ****=======**** I have read the author response. I have to clarify that my concerns is not on dynamics models, but on the dynamic model learning method. Although most of my concerns are responded, I still think that more comparative discussion with other dynamic model learning approaches are necessary.

[Author Response · NeurIPS 2019]

We wish to thank the reviewers for their time and thorough reviews. We deeply appreciate the unanimous appreciation of the clarity of the paper and the experiments that we curated to study in detail the strengths of our method, disentangling other possible sources of variation. We believe the main contributions of our work have been understood, but there are some points that we wish to explain in more detail, answering reviewers' comments and requests for clarification.

**Originality of the method and motivation for DAE.** As Reviewer 1 pointed out, regularization of trajectory optimization is a known technique in optimal control and we agree that we should have explained better the originality of our method and especially the motivation for using denoising autoencoders for regularization.

Our goal is to tackle high-dimensional problems (e.g. Ant has an observation space of 111 and action space of 8, we'll add this information in the Appendix) and expressive models of dynamics which not only means that regularization is needed, as we have demonstrated in the paper, but also that most other regularization techniques will fail. As Reviewer 1 points out, neural networks tend to fare better than many other techniques in modeling high-dimensional distributions. However, using a neural network or any other flexible parameterized model to estimate the input distribution poses a dilemma: the regularizing network which is supposed to keep planning from exploiting the inaccuracies of the dynamics model will itself have weaknesses which planning will then exploit. One is trying to patch the leak with something that leaks itself so let us call this the leaking patch problem.

During the early phases of the research which lead to the present paper, we were experimenting with different input density models including ensembles like those in Ref [20]. All of them suffered from the leaking patch problem. Only DAE avoids the problem and, to the best of our knowledge, nobody else has proposed using it for regularizing trajectories. Clearly DAE will also have inaccuracies but planning will not exploit them because unlike basically any other density models, DAE develops an explicit model of the *gradient* of logarithmic probability density.

How this keeps planning from exploiting the inaccuracies of DAE is easiest to see in gradient-based optimization. Let us assume that a density model (e.g. VAE) has a spurious maximum. If the gradients are calculated by backpropagation, the gradients will point towards this spurious maximum and planning will get there even if the spurious values will represent a fraction of the volume of the whole input space. With DAE, the gradients are obtained from the outputs of the network. There are spurious gradients but gradient directions will not correlate with the directions where the anomalies grow higher. This means that, theoretically, DAE should be a uniquely good solution for regularizing trajectory optimization, and the experimental results we presented support this. It is well known to be very hard to get gradient-based optimization to work with flexible models of dynamical systems and input densities because gradient ascent will immediately expose any inaccuracies of those models. Granted, we did not show that all the alternatives fail. Theoretical justification is nevertheless very important because it gives us some guarantees that the good results will extend to other problems, higher dimensions, and so on. Also, this justification is not limited to gradient-based optimization and indeed we showed that CEM benefits just as well.

**Previous works.** Works like [30, 31] use model-free RL, and use KL divergence of policies to prevent excessive change in output distribution. This is very different to what we do: we use a model to plan using a powerful optimizer, which can degrade performance if optimizing too much due to adversarial effect (Fig. 3). Similarly, [20] guides the evolution of a policy, while we learn a model that can solve novel goals at test time. [17] uses LTV models, which are faster but also generally too simple for many tasks (see [5]). With respect to (Gauss. Mix. Penalty for Traj. Optim. Problems), we scale to high dimensional problems, with more than 100 state variables.

**Use of CEM and re-implementation of PETS**. We use CEM for comparison with existing methods (PETS). We used a re-implementation of PETS only for the experiments in Section 4.2 to support gradient-based planning (we use the original implementation for all other comparisons). We will update the paper to show that the re-implementation produces the same results as the original implementation.

**Exploration.** The presented regularization method does penalize exploration and we could not come up with an exploration strategy that would lead to the state-of-the-art asymptotic performance. Experiments with simple exploration strategies (noise injection) show improved asymptotic performance (as we show in the appendix) but they do not reach the state-of-art. The proposed method can fall into a local optimum and this is what happens in our experiments because we do not reach the state-of-the-art asymptotic performance.

**Detailed questions by Reviewer 3**. 1) Please see the previous paragraph. 2) The transition function is deterministic in all the tasks we consider. 3) The results reported in Table 1 are obtained by evaluating trajectory optimization with different optimizers and regularizers using *pre-trained* dynamics models (that is, there is no learning process in these results). 4) We show that the DAE regularization is effective with both feed-forward (in mujoco experiments) and recurrent (in industrial process control experiments) neural network dynamics models. We used the same network architecture as PETS to avoid external sources of variations other than the DAE regularization. We are certainly very interested in extending our method to image observations using architectures inspired by the World Models paper, but tackling it in this work is out of scope.

[Meta-Review · NeurIPS 2019]

Reviewers find adding DAE style regularization in trajectory optimization phase of model-based RL interesting and appreciate the writing and execution of the paper. Reviewers though expressed concerns regarding the novelty of the work (a straightforward application of existing method) and would like to see more experiments demonstrating the effectiveness of proposed method under different dynamic models. Connection to behavior cloning and off-policy learning in model-free cases should be of interest to discuss. Overall, reviewers lean toward accepting the paper, we thus decided to accept it as is. Please address reviewers' comments in your final draft.